# Thiamine-Responsive Acute Pulmonary Hypertension of Early Infancy (TRAPHEI)—A Case Series and Clinical Review

**DOI:** 10.3390/children7110199

**Published:** 2020-10-28

**Authors:** Nalinikanta Panigrahy, Dinesh Kumar Chirla, Rakshay Shetty, Farhan A. R. Shaikh, Poddutoor Preetham Kumar, Rajeshwari Madappa, Anand Lingan, Satyan Lakshminrusimha

**Affiliations:** 1Rainbow Children’s Hospitals, Hyderabad 500034, India; dchirla@gmail.com (D.K.C.); farhanshaikh74@gmail.com (F.A.R.S.); 2Rainbow Children’s Hospital, Bengaluru 560037, India; rakshayshetty@gmail.com; 3Rainbow Children’s Hospital, Vikrampuri, Hyderabad 500009, India; drpreethamp@gmail.com; 4SIGMA Hospital, Mysore 570009, India; rajeesubbiah@yahoo.com; 5Narayana Multispecialty Hospital, Mysore 570019, India; anandlingan@gmail.com; 6Department of Pediatrics, UC Davis Children’s Hospital, Sacramento, CA 95817, USA; slakshmi@ucdavis.edu

**Keywords:** India, pulmonary hypertension, thiamine, beriberi, nitric oxide, sildenafil

## Abstract

Persistent pulmonary hypertension of the newborn (PPHN) is a syndrome of high pulmonary vascular resistance (PVR) commonly seen all over the world in the immediate newborn period. Several case reports from India have recently described severe pulmonary hypertension among infants in the postneonatal period. These cases typically present with respiratory distress in 1–6-month-old infants, breastfed by mothers on a polished rice-based diet. Predisposing factors include respiratory tract infection such as acute laryngotracheobronchitis with change in voice, leading to pulmonary hypertension, right atrial and ventricular dilation, pulmonary edema and hepatomegaly. Mortality is high without specific therapy. Respiratory support, pulmonary vasodilator therapy, inotropes, diuretics and thiamine infusion have improved the outcome of these infants. This review outlines four typical patients with thiamine-responsive acute pulmonary hypertension of early infancy (TRAPHEI) due to thiamine deficiency and discusses pathophysiology, clinical features, diagnostic criteria and therapeutic options.

## 1. Introduction

Pediatric pulmonary hypertension (PH) is most commonly associated with cardiac and pulmonary diseases. Among less common causes, various hematologic, hepatic, metabolic, oncologic, genetic, and rheumatologic disorders are described in the literature [1]. The presentation of Infantile PH in the setting of vitamin deficiencies is exceedingly rare, though associations with deficiencies in vitamin D, thiamine, and vitamin C are reported [2,3,4]. Congestive cardiac failure in infants from thiamine deficiency is well documented in some specific geographic locations. In acute wet beriberi or Shoshin beriberi, rapid deterioration in cardiac function leads to hypotension, pulmonary edema, cardiomegaly and high lactate levels. However, in the past, PH was not reported in these infants [5].

Persistent pulmonary hypertension of the newborn (PPHN) presents in the neonatal period with an incidence of 2 per 1000 live births. Early infancy has traditionally been an uncommon age for presentation with new-onset PH. However, recently, several cases of severe PH due to thiamine deficiency were reported during early infancy from India. Here we review clinical presentation, hospital course, treatment options and outcome of four cases of thiamine-responsive acute pulmonary hypertension of early infancy (TRAPHEI). We do not believe this is a new disease entity. However, the easy availability of point of care echocardiography has made the diagnosis of PH easier in outpatient, inpatient wards and intensive care settings [2]. During initial recognition of this condition, thiamine deficiency was not recognized in many instances resulting in high mortality. A high index of suspicion for early identification of thiamine deficiency as a cause of TRAPHEI in early infancy and a low threshold for thiamine administration are keys to the optimal management of these critically ill infants and to reducing their morbidity and mortality.

## 2. Case Reports

Details of four cases of TRAPHEI are listed in Table 1. Case 1 was a two-month-old infant with worsening respiratory distress, fever, cough, runny nose for 5 days. Chest radiograph showed minimal cardiomegaly (Figure 1A). Despite therapy with dexamethasone, the infant deteriorated with increasing respiratory distress, hypoxemia and bradycardia requiring conventional mechanical ventilation and subsequently high frequency ventilation (HFV). Echocardiography showed marked right ventricular dilation and flattening of the inter-ventricular septum (Figure 1B), severe tricuspid regurgitation jet (Figure 1C), pulmonary arterial acceleration time (PAAT) of 59 msec and tricuspid annular plane systolic excursion (TAPSE) of 8.0 mm were observed. The left ventricular ejection fraction (LVEF) was 58%. Despite inhaled nitric oxide (iNO) and milrinone, the patient remained acidotic with a pH of 7.23, a lactate level of 6.3 mmol/L and a PaO_2_ of 44 mmHg with FIO_2_ of 1.0. 

The second case was a 3½-month-old infant with respiratory distress. The chest radiograph was normal. This infant rapidly deteriorated with acidosis and hypoxemia, requiring conventional mechanical ventilation but later switched to HFV with pulmonary vasodilator and inotropic support. Echocardiography showed marked right ventricular and right atrial dilation (Figure 2) and a right-to-left shunt at the patent foramen ovale (PFO) with the interventricular septum deviated to the left. Moderate tricuspid regurgitation, a PAAT of 54 msec and TAPSE of 7.0 mm were observed.

The third patient was a two-month-old boy with respiratory arrest at a community healthcare facility requiring mechanical ventilation and vasopressor support. In the next 12 h, the infant deteriorated further with increasing respiratory distress and hypoxemia and required HFV. The infant was treated with enteral sildenafil and bosentan with no improvement in hypoxemia (PaO_2_: 43 mmHg with FIO_2_ of 1) or lactic acidosis (8.1 mmol/L).

The fourth patient was a 3-month-old infant brought to the hospital with a history of fever for 4 days and grunting for a few hours. On examination, this baby was cold, lethargic, gasping with cyanosis requiring intubation and resuscitation with IV normal saline, dopamine, and sodium bicarbonate. Echocardiogram revealed dilated right atrium, right ventricle with bulging of the interventricular septum to the left with tricuspid regurgitation.

In all four cases, mothers were on a diet of washed, polished white rice. Infants were exclusively breastfed. These infants received an IV dose of thiamine (100 mg) over 30 min followed by parenteral and then enteral supplementation with resolution of symptoms (Table 1). Repeat echocardiogram showed resolution of PH and infants were healthy at follow-up. Mothers received dietary recommendations and oral thiamine supplementation. The metabolic screen for inborn errors of metabolism was negative in all cases.

## 3. Review of Literature

### 3.1. Clinical Presentations

PPHN is a well-known critical illness in the neonatal period. Beyond the neonatal period, during early infancy, PH is mainly caused by cardiac and pulmonary diseases such as congenital heart disease, bronchopulmonary dysplasia (BPD) or congenital diaphragmatic hernia (CDH). Infantile PH is uncommon in the US and UK in the absence of pre-existing cardiac or pulmonary disease. [6] There has been a rapid increase in the recognition of infants with PH in India that respond to thiamine [7].

While malnutrition-induced PH is likely a rare finding, instances of vitamin D3, vitamin C and thiamine (B1) deficiency causing PH are observed across different age groups. Recently several cases of postneonatal pulmonary hypertension are reported from India. Infantile thiamine deficiency related symptoms most commonly occur between 1 and 7 months of age. However, acute cardiac manifestations of Shoshin beriberi are more common among young infants between 1 and 3 months of age [8]. Risk factors for TRAPHEI include exclusive breastfeeding by thiamine-deficient but otherwise asymptomatic mothers. Mothers who consume exclusive polished white rice that has been washed multiple times are at the highest risk of thiamine deficiency. Young infants (1–3 months of age) exclusively breastfed are at greatest risk for developing thiamine deficiency. [9] Thiamine deficiency has a wide range of clinical presentations, with high fatality in untreated cases and survivors usually have long-term sequelae. Although thiamine deficiency is effectively treatable, it continues to affect infants in both developed and low-to-middle income countries (LMIC) with potentially serious and life-threatening consequences [10,11,12].

Thiamine deficiency has a wide clinical spectrum and is typically described as three classic presentations: wet, Shoshin and dry beriberi, and as such it is frequently missed in infancy. The word beriberi is derived from a Sinhalese word referring to extreme weakness. Shoshin is derived from Japanese (sho–acute damage and shin–heart). The infants presented in this case series resemble Shoshin beriberi. Three clinical forms have been identified in infants: acute severe cardiac, aphonic form, and pseudo-meningitic forms. Infants have an inherently greater thiamine requirement due to their relatively high metabolic rates [13]. Differential diagnosis of thiamine deficiency in infants include sepsis, meningoencephalitis, cardiomyopathy, complicated falciparum malaria, infantile kwashiorkor, metabolic encephalopathy and idiopathic pulmonary arterial hypertension. The clinical presentation in our patients was typical of PH exacerbated by a respiratory illness in previously healthy infants on exclusive breast feeding. These patients had severe respiratory distress, refractory hypoxemia, PH and lactic acidosis with some requiring mechanical ventilation, pulmonary vasodilators and inotropic support. These infants had a typical history of maternal customary dietary restriction and consumption of exclusive repeatedly washed and polished white rice and wheat flour. Infants rapidly responded to IV infusion of thiamine 100 mg over 20–30 min and showed clinical improvement over the next 2–4 h. All infants received enteral thiamine for a few weeks. After 24–48 h of thiamine, significant improvement in PH was observed in these infants. Although, sildenafil and bosentan were used in the management of these patients, these medications did not result in any significant change in clinical outcomes. Follow-up after 4 to 6 weeks of thiamine therapy showed complete resolution of this form of postneonatal pulmonary hypertension.

Thiamine in free form occurs in plasma and represents only a small part of the whole-body thiamine; whereas its phosphate ester, thiamine diphosphate (TDP) is the predominant intracellular metabolite. TDP levels provide a better measure of body thiamine status but do not assess thiamine metabolic function [13,14]. Erythrocyte transketolase activity (ETKA) is more accurate in assessing the functional thiamine status of the body [13]. ETKA is considered as the most accurate biochemical marker to diagnose thiamine deficiency, however it is expensive, time consuming and not readily available in resource-limited settings. We found low plasma levels of thiamine in our patients along with their mothers. Rapid clinical improvement to thiamine challenge and reversal of PH in these infants is more useful in diagnosing thiamine deficiency as a cause of TRAPHEI in our clinical settings. Risk factors and clinical features of TRAPHEI caused by thiamine deficiency are described below in Figure 3 and Figure 4.

### 3.2. Indian Experience

A recent estimation of the prevalence of the population at risk of dietary thiamine inadequacy in India (based on national food balance sheets) was 14.8% [15]. Though studies on infantile thiamine deficiency are reported from South Asian and African countries, most cases of TRAPHEI have been reported from different parts of India. There are a few case series of PH due to thiamine deficiency reported from Kashmir, northern state of India and Telangana and Karnataka in southern India. In a descriptive study of twenty-three infants from Srinagar, Kashmir state in India who presented with acute life-threatening metabolic acidosis (blood pH < 7.0) due to thiamine deficiency, only four infants required ventilation for a brief period. On admission, twelve (52%) infants presented with cardiogenic shock but authors did not find any case of hypoxemic respiratory failure in this group [3]. In another recent publication from the same group, Bhat et al. studied 29 infants with a mean postnatal age of 78.45 days, who presented with clinical signs of acute onset PH [16]. All the babies included were exclusively breastfed and are from middle and lower socio-economic class families. Right heart failure and acute metabolic acidosis were universal findings. Thiamine levels were sent for only two infants due to financial constraints, but most of them had a prompt response to thiamine. A total of 12 infants needed mechanical ventilation with median duration of 8 h. Four infants died due to irreversible shock and renal failure. There was a statistically significant drop in pulmonary pressures on repeat echocardiography, seen in 25 babies who were followed up after 4–6 weeks [16]. A case series by Narsimha et al. described 55 infants with mean age of 3.9 months with clinical features of high-output cardiac failure with PH. Mean duration of illness was 7.5 days but data regarding severity of illness and need for ventilation were not provided. All these infants and their mothers belonged to low socio-economic groups and were thiamine-deficient based on low levels of ETKA. These infants showed a rapid response to thiamine confirming the diagnosis of cardiac beriberi. Nineteen babies were followed up after 2–3 weeks and demonstrated the resolution of PH with a reversal of echocardiographic abnormalities [17]. A similar case was reported from West Africa: a 3-month-old infant presented with severe pneumonia, progressing rapidly into respiratory failure. Due to limitation of testing and ventilation capabilities in this setting, the baby was given a trial of thiamine as a last resort. The baby improved dramatically within 12 h, was off oxygen in 48 h and was discharged on the fifth day [18].

### 3.3. Pathophysiology

There is no endogenous thiamine synthesis in humans and the body’s requirements depend exclusively on dietary supply. The combination of limited body storage and a high turnover rate (half-life < 10 days) results in potential depletion of thiamine stores within 2 weeks if it is not continuously replaced. Exclusively breastfed infants with their high metabolic demand and low thiamine level in mothers are at the highest risk. Symptoms are sometimes precipitated with the presence of co morbidity such as sepsis and prolonged glucose infusion. Thiamine (Vitamin B1) is an essential micronutrient with dual coenzymatic and non-coenzymatic functions. The biologically active form of thiamine, thiamine pyrophosphate, is a cofactor for enzyme pyruvate dehydrogenase, a glycolytic enzyme that catabolises the decarboxylation of pyruvate to acetyl coenzyme A. Deficiency of thiamine therefore results in a metabolic block in the last step of glycolysis and excess pyruvate gets converted to lactate. Thus, thiamine deficiency leads to an acquired mitochondrial disorder (Figure 4). Pulmonary hypertension in cardiac beriberi is rare. We speculate two mechanisms for PH in these infants with thiamine deficiency. The first mechanism is that of pulmonary venous hypertension. It occurs as a result of progressive energy failure and/or damage to the myocardium that leads to circulatory breakdown, lactic acidosis and left ventricular dysfunction leading to elevated left ventricular end diastolic pressure. Direct impairment of myocardial energy production has been proposed as one possible mechanism as thiamine is required as a cofactor for energy production [19]. Biventricular dysfunction has been observed in thiamine deficiency, but right ventricle is more profoundly affected than left ventricle [20]. Lactic acidosis can contribute to acute pulmonary vasoconstriction in our critically ill patients even after correction of hypoxemia [21]. The second mechanism is through the production of superoxide anions and other reactive oxygen species (ROS) and reactive nitrogen species (RNS). Thiamine deficiency is associated with increased production of ROS [22]. Superoxide anions can inactivate nitric oxide producing peroxynitrite [7,8], a powerful vasoconstrictor that can exacerbate PH (Figure 4).

Interestingly, all patients described in our case series are male infants. A review of literature of cases of infantile PH from India show that 65–70% of infants reported to date are male infants [7,23,24]. It is not clear if gender plays a role in the pathogenesis of PH. We cannot rule out societal preference to seek care for male infants. In addition, in some of these cases, there is past history of a sibling suffering from a similar illness often resulting in death. Familial recurrence could suggest the persistence of thiamine deficiency in the mother from pregnancy to pregnancy. A genetic component increasing susceptibility to low thiamine levels cannot be ruled out (Figure 4).

### 3.4. Treatment

General management guidelines for TRAPHEI are shown in Figure 5. Stabilizing airway, breathing and circulation followed by obtaining blood gas, lactate, thiamine level, chest X-ray and an echocardiogram are important. In our case series, we used two-dimensional echocardiography for the diagnosis of PH. Cardiac catheterization was not done in any of our infants with TRAPHEI. Though echocardiography overestimates pulmonary hypertension, a doppler-assessed PA systolic pressure of >40 mm Hg was arbitrarily considered as a cut-off value for our infants with TRAPHEI. In these cases of refractory hypoxemia and persistent lactic acidosis, the optimal approach is to exercise a high index of clinical suspicion and early therapeutic thiamine challenge which can help the rapid resolution of symptoms. Once patients were clinically suspected of thiamine deficiency as a cause of TRAPHEI, supportive measures such as mechanical ventilation, pulmonary vasodilators and circulatory support were optimized. Once a blood sample was drawn for measurement of thiamine level, a dose of 100 mg of thiamine diluted in normal saline was given as an IV infusion over 20–30 min. Thiamine challenge reversed biochemical abnormalities and cardiorespiratory status in our patients within 2–4 h. We continued thiamine infusion 100 mg per day as infusion for 2–3 days and later both infant and mother were placed on enteral thiamine supplementation for a few months. There is no conclusive evidence regarding pediatric thiamine dosage for severe acute cardiac illness. In adult patients, authors described doses of thiamine in a range of 50–1500 mg per day and a duration of 7 days until discharge [25]. Rao and Chandak used 150 mg IV thiamine to treat breastfed infants under 6 months of age presenting with cardiac failure and/or tachypnea, whereas Bhatt et al. successfully treated using 100 mg of thiamine daily. Hubert Barennes et al. treated infants with acute symptomatic thiamine deficiency with an intramuscular (IM) or slow IV injection of 50 mg thiamine [10,14,15]. All infants with pulmonary hypertension completely resolved at 4 weeks of follow-up and no recurrence was observed at 6 months of follow-up.

## 4. Conclusions

Diagnosis of TRAPHEI can be considered if the infant at risk presents with respiratory failure and has all of the following criteria: (1) newly diagnosed pulmonary hypertension without any pre-existing congenital heart disease or chronic pulmonary condition, (2) metabolic acidosis with elevated lactate, (3) no other cause for this illness including sepsis, (4) maternal dietary history suggestive of thiamine-deficiency in an exclusively or predominantly breastfed infant and (5) rapid response to IV thiamine challenge.

Early recognition and diagnosis of thiamine deficiency is imperative as specific therapy decreases morbidity and mortality. These cases described here highlight a potentially critical but reversible manifestation of thiamine deficiency. In the absence of specific diagnostic tests in resource-limited settings, a low threshold for a therapeutic thiamine challenge is an easier way to diagnose thiamine deficiency. At the community level, improvised strategies like programmatic approaches to rice and salt fortification, thiamine supplementation to pregnant and lactating mothers, and dietary modification like the use of parboiled rice might improve thiamine level in exclusively breastfed infants. Awareness, education and training of health care providers about TRAPHEI in infants can help in early recognition and treatment of this life-threatening condition. Future investigations into the link between thiamine (B1) deficiency and TRAPHEI may provide insights for the understanding of the pathophysiology, diagnostic criteria, appropriate preventive and therapeutic strategies relevant to the condition.

## Figures and Tables

**Figure 1 children-07-00199-f001:**
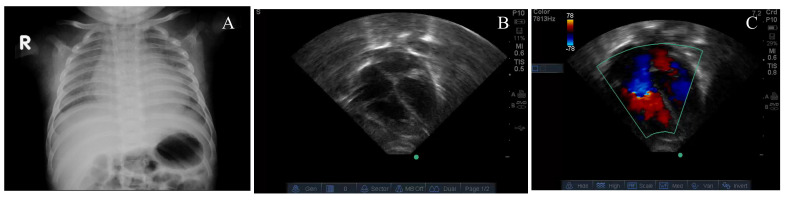
Chest X-ray showing mild cardiomegaly (**A**), four-chamber view of the echocardiogram showing septal flattening (**B**) and color Doppler showing tricuspid regurgitation (**C**) from case # 1.

**Figure 2 children-07-00199-f002:**
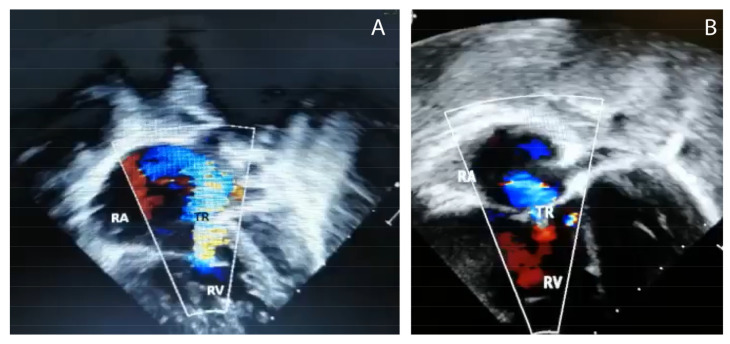
Echocardiography with Doppler images from case # 2 showing (**A**) hypertrophic right ventricle (RV) and dilated right atrium (RA), and (**B**) tricuspid regurgitation (TR),

**Figure 3 children-07-00199-f003:**
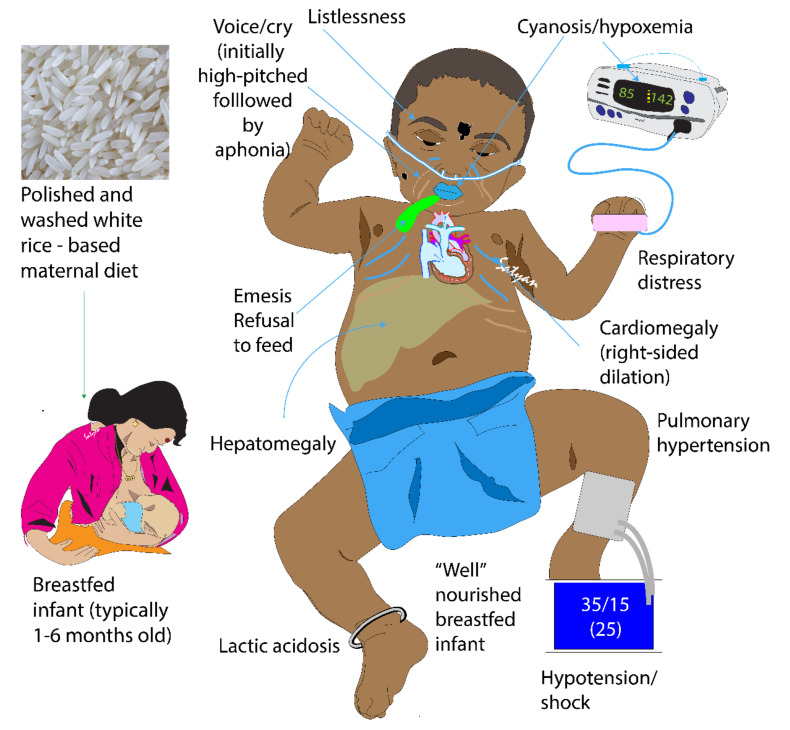
Clinical Features of thiamine-responsive acute pulmonary hypertension in early infancy (TRAPHEI) (copyright Satyan Lakshminrusimha).

**Figure 4 children-07-00199-f004:**
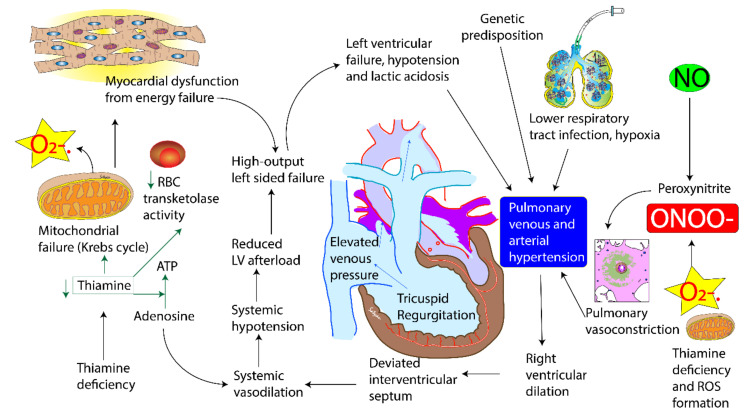
Pathophysiology of thiamine deficiency and pulmonary hypertension of early infancy. (copyright Satyan Lakshminrusimha).

**Figure 5 children-07-00199-f005:**
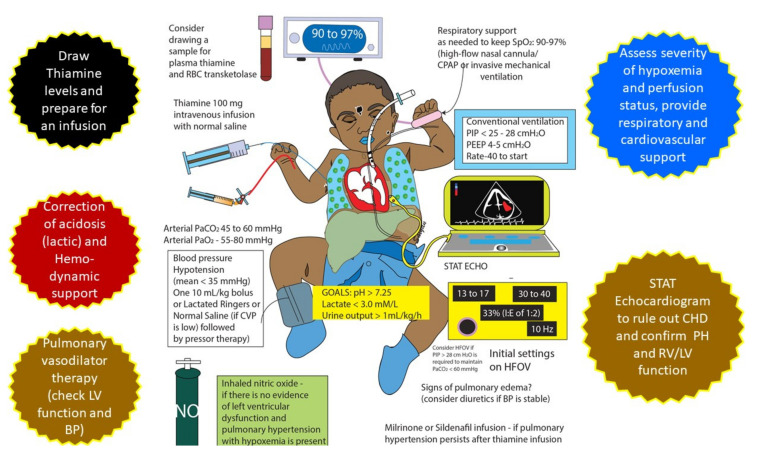
Treatment protocol for TRAPHEI (copyright Satyan Lakshminrusimha).

**Table 1 children-07-00199-t001:** Clinical presentation of four cases of thiamine-responsive acute pulmonary hypertension of early infancy (TRAPHEI).

	Case 1	Case 2	Case 3	Case 4
Age at presentation	2 months	3½ months	2 months	3 months
Gender	Male	Male	Male	Male
Birth weight	3.2 kg	3.0 kg	2.9 kg	2.5 kg
Current weight	4.7 kg	5.6 kg	4.1 kg	4.7 kg
Consanguinity	No	No	Yes	Yes
Family history of similar illness	-	-	Brother died at 3 months	Sister died at 3½ months
Initial symptoms and signs	fever, cough, runny nose, respiratory distress, stridor, and irritability	feeding difficulty, cough and worsening respiratory distress	respiratory arrest and intubated in community hospital	fever, lethargy, respiratory distress
Maternal/infant diet	Polished, washed, white rice/exclusive breastfeeds	Polished, washed, white rice/exclusive breastfeeds	Polished, washed, white rice/exclusive breastfeeds	Polished, washed, white rice/exclusive breastfeeds
Hepatomegaly (cm below costal margin)	3 cm	2 cm	2 cm	3 cm
Worst pH	6.98	7.08	7.09	6.92
Highest lactate (mM/L)	9.8	6.8	8.5	16.4
Lowest PaO_2_/FiO_2_	40 mmHg	65 mmHg	35 mmHg	55 mmHg
Peak estimate of RVSP (mm Hg)	82	74	60	67
Ventilation mode High frequency ventilation (HFV) or Conventional mechanical ventilation (CMV)	HFV	HFV	HFV	CMV
Pulmonary vasodilators	Nitric oxide (NO)	NO Sildenafil	Sildenafil Bosentan	-
Circulatory support	Milrinone Epinephrine Vasopressin Norepinephrine	Epinephrine Milrinone	Dopamine	Dopamine Furosemide
Thiamine level (infant/mother) (normal—66.5 to 200 nM/L)	10 (infant)	12.46 (infant) 2.38 (mother)	3.7 (infant) 4.2 (mother)	75 (mother)
Response to thiamine (hours or days after first infusion)				
Extubation	3 days	40 h	34 h	36 h
Wean to room air	5 days	4 days	4 days	3 days
Resolution of PH by echocardiogram after thiamine	Improvement started by 4 h, resolution by 5 days.	Improvement by 3 h, resolution by 6 days.	Improvement started by 8 h, resolution by 7 days.	Resolution 5 days.

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
