# Peer review of "Thiamine-Responsive Acute Pulmonary Hypertension of Early Infancy (TRAPHEI)—A Case Series and Clinical Review"

_children, 2020, doi:10.3390/children7110199_

Round 1

Reviewer 1 Report

The authors are describing a case series of pulmonary hypertension in Indian infants that are caused by thiamine deficiency.   As the authors had mentioned in their manuscript, this presentation of thiamine deficiency is not a traditional known cause although it had been described by previous authors from the Indian sub-continent. This clinical problem is more prevalent in society whose staple diet is predominantly well milled rice causing thiamine deficiency.  These infants who are exclusively breast fed are therefore similarly exposed to the risk of thiamine deficiency.  The authors had described four patients in great details documenting the presentation, clinical findings, laboratory findings and response to treatment.  The authors also highlight the importance of recognition of thiamine deficiency as an important cause of these type of infantile pulmonary hypertension that have serious consequences or result in death if treated late or left untreated.  They have documented excellent response to treatment with thiamine with complete resolution of pulmonary hypertension.

I would suggest to the authors to consider enhancing the manuscript by revising it to include the following suggestions: -

  1. Clarify the title by including a subtitle to highlight thiamine deficiency as the cause of this reversible pulmonary hypertension. This is in contrast to their use of “Indian post-neonatal
  2. Include in their introduction, the traditional manner of presentation of beri-beri and perhaps some historical perspective to include the more serious Soshin type. Then introduce their reason for including these four patients as a well-documented series with their supposition that this is a “new” presentation.
  3. Shorten the case reports to be more succinct to highlight the clinical picture. Here, I would anticipate inclusion on clinical features of cardiac output (low versus high) and hepatic function. A table of the salient clinical data will enhance the manuscript.
  4. I would also like the authors to discuss their use of bosentan if it has important bearing in management of pulmonary hypertension in these patients.

Author Response

We would like to thank Editor and both Reviewers for their thoughtful comments and inputs. Please find attached  point to point response to reviewers comments .

The authors are describing a case series of pulmonary hypertension in Indian infants that are caused by thiamine deficiency.   As the authors had mentioned in their manuscript, this presentation of thiamine deficiency is not a traditional known cause although it had been described by previous authors from the Indian sub-continent. This clinical problem is more prevalent in society whose staple diet is predominantly well milled rice causing thiamine deficiency.  These infants who are exclusively breast fed are therefore similarly exposed to the risk of thiamine deficiency.  The authors had described four patients in great details documenting the presentation, clinical findings, laboratory findings and response to treatment.  The authors also highlight the importance of recognition of thiamine deficiency as an important cause of these type of infantile pulmonary hypertension that have serious consequences or result in death if treated late or left untreated.  They have documented excellent response to treatment with thiamine with complete resolution of pulmonary hypertension.

I would suggest to the authors to consider enhancing the manuscript by revising it to include the following suggestions: -

We thank the reviewer for his/her valuable suggestions and input to improve our manuscript.

  1. Clarify the title by including a subtitle to highlight thiamine deficiency as the cause of this reversible pulmonary hypertension. This is in contrast to their use of “Indian post-neonatal

             Response : We thank the reviewer for reminding us to emphasize thiamine deficiency and response in this condition. We have appropriately changed the title of this review to  “Thiamine Responsive Acute Pulmonary Hypertension of Early Infancy (TRAPHEI)”.

  1. Include in their introduction, the traditional manner of presentation of beri-beri and perhaps some historical perspective to include the more serious Soshin type. Then introduce their reason for including these four patients as a well-documented series with their supposition that this is a “new” presentation.

          Response: In lines 40,41and 42 we have included a brief description of presentation of shoshin beriberi. However, pulmonary hypertension was not commonly reported in traditional shoshin beriberi until recent reports highlighted PH by Bhat et al in reference # 16.

  1. Shorten the case reports to be more succinct to highlight the clinical picture. Here, I would anticipate inclusion on clinical features of cardiac output (low versus high) and hepatic function. A table of the salient clinical data will enhance the manuscript.

Response: We thank you for this suggestion. We have made a summary of clinical presentations and outcome of four cases in Table 1. We also shortened the description of our cases of TRAPHEI.

  1. I would also like the authors to discuss their use of bosentan if it has important bearing in management of pulmonary hypertension in these patients.

Response: We do not believe that bosentan had any impact on outcome and do not encourage its use in this condition (unless patients do not respond to thiamine). We have emphasized this point in line 257 and 258.

We thank both reviewers and editor for their comments and suggestions.

Best Regards

Nalinikanta Panigrahy

Satyan Lakshminrusimha

Reviewer 2 Report

The review nicely summarizes an important, often overlooked etiology of post-natally acquired pulmonary hypertension.  The cases are clearly presented, and the Discussion is largely on-target, and tailored to the readership.  The Discussion about the putative mechanisms is well done.

The manuscript would benefit by a more careful copyediting by a fluent English speaker.  There are numerous cases of omitted articles (a, the).  Contractions should be fully spelled out: “did not” instead of “didn’t,” for example.

Figure 3 can be omitted, since it does not add anything beyond the text.

Figure 4: the images of the rice and mother ought to be omitted from the diagram.  This will render the figure less crowded appearing.  The ideas of the diagram are the mechanisms by which thiamine deficiency—from any etiology, not just a particular maternal dietary deficiency like the polished rice—contributes to the downstream pathophysiologic events. 

Figure 5 and Table 1 are redundant.  It is very clear (throughout the manuscript) that thiamine deficiency should be suspected in such clinical settings, and that the fastest, safest way to make the diagnosis is to treat with thiamine, a relatively low-cost approach.  The remainder of the information conveyed in Figure 5 and Table 1 are the standard approaches to pulmonary hypertension, none of which are specific to this particular patient category.  “Brevity is the soul of wit,”  Alexander Pope.

Author Response

The review nicely summarizes an important, often overlooked etiology of post-natally acquired pulmonary hypertension.  The cases are clearly presented, and the Discussion is largely on-target, and tailored to the readership.  The Discussion about the putative mechanisms is well done.

Response: We thank the reviewer for his/her constructive criticism and have made extensive edits to the manuscript.

The manuscript would benefit by a more careful copyediting by a fluent English speaker.  There are numerous cases of omitted articles (a, the).  Contractions should be fully spelled out: “did not” instead of “didn’t,” for example.

We made several corrections in grammar and syntax. The paper was reviewed by Ms. Lauren E. Hazewski, an editorial assistant with extensive experience in manuscript editing.

Figure 3 can be omitted, since it does not add anything beyond the text.

Response: We agree with the reviewer that brevity is important and have reduced the paper by over 1000 words and removed one table. However, we believe the revised Fig 3 plays a key role in emphasizing important aspects of thiamine responsive pulmonary hypertension. We hope that including this figure will enhance citation of this paper and provide an educational tool for trainees enabling early recognition of this condition. We request the reviewer to permit us to retain this figure.

Figure 4: the images of the rice and mother ought to be omitted from the diagram.  This will render the figure less crowded appearing.  The ideas of the diagram are the mechanisms by which thiamine deficiency—from any etiology, not just a particular maternal dietary deficiency like the polished rice—contributes to the downstream pathophysiologic events. 

Response: We agree with the reviewer and removed maternal diet from Fig 4. We added it to clinical features in figure 3 as this history provides an important clue to diagnosis.

Figure 5 and Table 1 are redundant.  It is very clear (throughout the manuscript) that thiamine deficiency should be suspected in such clinical settings, and that the fastest, safest way to make the diagnosis is to treat with thiamine, a relatively low-cost approach.  The remainder of the information conveyed in Figure 5 and Table 1 are the standard approaches to pulmonary hypertension, none of which are specific to this particular patient category.  “Brevity is the soul of wit,”  Alexander Pope.

Response: We removed this table completely. We would prefer to retain figure 5 as a teaching tool to provide a synopsis of management of this life-threatening condition. Hundreds of infants have lost and continue to lose their life to this preventable condition. We intend to enhance education on early diagnosis and treatment with this low-cost intervention as mentioned by the reviewer.

We thank both reviewers and editor for their comments and suggestions.

Best Regards

 Nalinikanta Panigrahy

Satyan Lakshminrusimha

Round 2

Reviewer 1 Report

The manuscript reads better and the message for thiamine responsive pulmonary hypertension in infants is well highlighted and readers would learn from their reperience.

i would recommend to add a short subtitle such as:  "A case series and clinical review". 

There are some minor spelling errors: Line 219 in their table, Improvent should be improvement

Author Response

Dear Editor 

We are thankful again and appreciate  the reviewer for his/her kind words and thoughtful inputs .

We Thank Editorial team for All your Supports

Reviewer Comments :

The manuscript reads better and the message for thiamine responsive pulmonary hypertension in infants is well highlighted and readers would learn from their reperience.

i would recommend to add a short subtitle such as:  "A case series and clinical review". 

RESPONSE : Thanks for this suggestion.We have Added a subtitle "- A case series and clinical review" as per Reviewer suggestion

There are some minor spelling errors: Line 219 in their table, Improvent should be improvement

RESPONSE : Thanks Again for this inputs . Spelling checks done and corrected accordingly.

Best Regards 

Nalinikanta Panigrahy

Satyan Lakshminrushimha